Matters arising

# Increase of P-wave velocity due to melt in the mantle at the Gakkel Ridge

Zhiteng Yu [1,2,3] ✉ & Satish C. Singh [2]

ARISING FROM I. Koulakov et al. *Nature Communications* https://doi.org/10.1038/s41467-022-30797-4 (2022)

Microearthquake and seismic tomography studies are commonly used to apprehend melt accumulation, melt migration and submarine eruptions at mid-ocean ridges. Koulakov et al.[1] observe deep microseismicity, accompanied by an increase in P-wave velocities (Vp), decrease in S-wave velocities (Vs), leading to high Vp/Vs ratios at 0–13 km below the seafloor at the Gakkel Ridge, which they interpret to be due to the presence of melt in the mantle. However, the presence of melt in the mantle will decrease, not increase, the P-wave velocity. The reanalysis of the picked arrival times indicates that the high Vp/Vs ratios obtained by Koulakov et al.[1] result from misidentification of seismic phases, and hence their interpretation of a low degree of melting in the mantle is questionable.

Seismic velocities (compressional P-wave and shear S-wave (Vp, Vs)) can be used to determine rock types in the Earth and to help understand different physical and chemical processes within the Earth. In the recently published paper, Koulakov et al.[1] use microearthquake data to determine the Vp and Vs beneath axial volcanoes of the ultraslow-spreading Gakkel Ridge at 85°E. They find an increase in P-wave velocities and a decrease in S-wave velocities from the seafloor down to 13 km depth, leading to high Vp/Vs ratios, which they interpret to be caused by fluid-saturated fractured rocks from the seafloor down to 5 km depth, and by a stable magma reservoir at 5–13 km. However, when cracks are saturated with fluids in the crust, both Vp and Vs would be reduced but Vp/Vs ratio would increase as Vs decreases more than Vp[2,3] due to the presence of fluids. In the mantle, high Vp/Vs ratios can be due to either the presence of melt[4], or a high degree of serpentinization[5,6], or fractured mantle rocks with fluid-filled veins[3]. However, in all these cases, both Vp and Vs would be reduced[2-6], contrary to the high Vp and low Vs anomalies observed by Koulakov et al.[1]. Koulakov et al.[1] argue that the high Vp and low Vs anomalies are commonly observed beneath volcanoes on land, which are interpreted to be due to the presence of magma, however, such an interpretation cannot be valid in the oceanic domain, especially in the mantle, because the Vp of mantle peridotite (~8 km/s) is much higher than the Vp of melt or other fluids ($H_2O$, $CO_2$) (1.0–3.5 km/s)[7] or frozen gabbroic sills (~7 km/s)[8], and therefore, the effective Vp of a composite rock would always be less than the surrounding mantle rocks.

To demonstrate that the increase of P-wave velocity in the presence of fluid in the mantle is nonphysical, we used a differential effective medium theory[4] to compute the mantle Vp and Vs with different aspect ratios and velocities for ellipsoidal melt inclusions (Fig. 1a, b). In the starting one-dimensional (1D) velocity model of Koulakov et al.[1], the mantle Vp and Vs at 10 km depth are 7.42 and 4.29 km/s, respectively (Fig. 1a, b) which become Vp = 7.87 km/s (+6%) and Vs = 4.03 km/s (−6%)[1] after the inversion (Fig. 1a, b). If we assume that the mantle Vs decrease (−0.26 km/s) is due to spherical melt inclusions (aspect ratio $r = 1$), a maximum of ~7.0% of melt fraction would be required to explain this decrease (Fig. 1a). For higher aspect ratios ($r = 2$–100), the amount of melt would be much less, ranging from ~0.2 to 6% (Fig. 1a). Assuming various aspect ratios ($r = 1$–100), an increase in mantle Vp (+0.45 km/s) would require a melt fraction of 9.4–24.2% but Vp of the melt inclusion has to be 10–100 km/s (Fig. 1b), which is nonphysical, and hence the high Vp anomaly at 10 km depth observed by Koulakov et al.[1] cannot be due to the presence of melt.

We also analyzed the reliability of the picked P- and S-wave phases. The seismic stations were deployed on the ice floes[1]. As S-waves cannot travel in the water, the Vs tomographic model was obtained using sP-wave phases[1], which are S-waves converted to P-waves on the sea bottom. Assuming that both the P-wave and sP-waves have the same ray paths, their time differences ($t_S - t_P$) would be only caused by different Vp and Vs in the subsurface, which is the basic principle in the inversion method used by Koulakov et al.[1]. We counted time differences on each station (Fig. 1c), and found that 80% of the time differences are less than 2.5 s (Fig. 1c). The hypocentral distance (D) could be approximated using a simple equation[9]:

$$D\ (\text{km}) = (t_S - t_P) \frac{VpVs}{Vp - Vs} \tag{1}$$

In the starting 1D model[1], Vp increases from 4.8 km/s at 1.7 km to 7.8 km/s at 30 km depth with a Vp/Vs ratio of 1.73; therefore, $\frac{VpVs}{Vp-Vs}$ would be 6.5–10.6, resulting in $D$ of 16.25–26.5 km assuming a time difference of 2.5 s. The estimated epicentral distances would be ≤20 km, which is too small for the large (~70 km × 40 km) deployed

[1]Geo-Ocean, Univ Brest, CNRS, Ifremer, UMR6538, F-29280 Plouzané, France. [2]Laboratoire de Géosciences Marines, Institut de Physique du Globe de Paris, Université de Paris Cité, 75005 Paris, France. [3]Key Laboratory of Submarine Geosciences, Second Institute of Oceanography, Ministry of Natural Resources, 310012 Hangzhou, China. ✉e-mail: ztyu@sio.org.cn

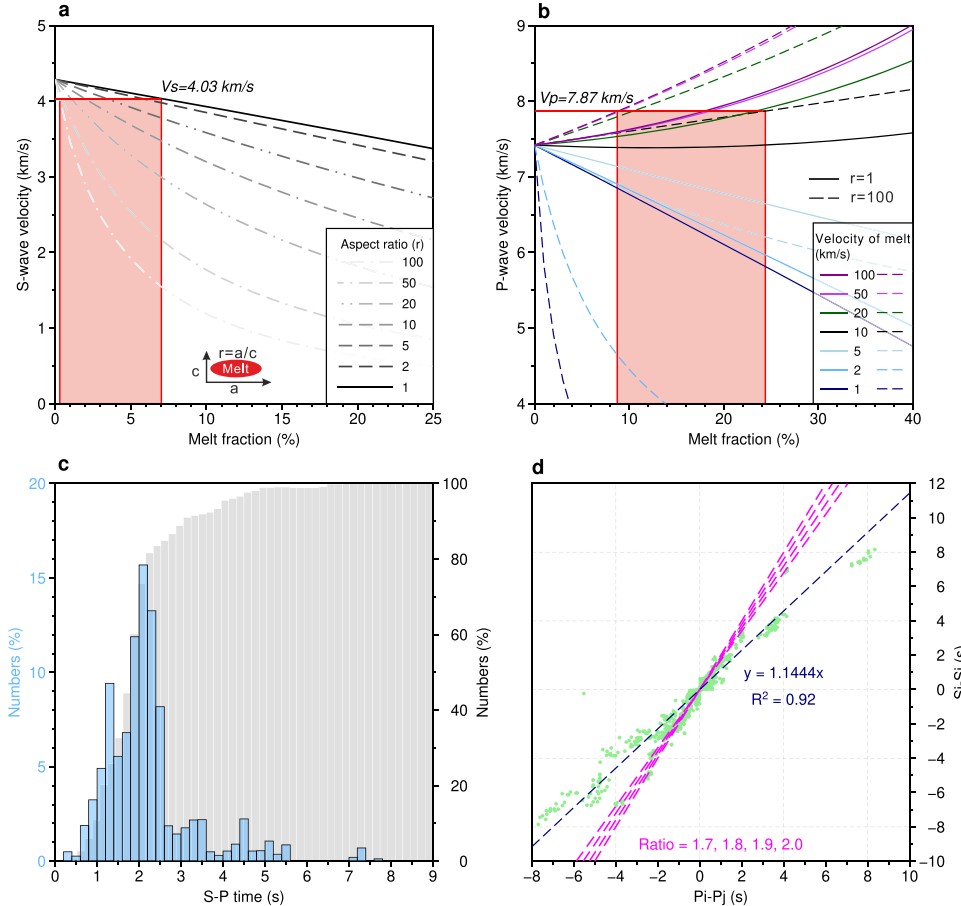

**Fig. 1 | Analyses of the inverted velocities and picked phases by Koulakov et al.[1].** **a** Variation in the S-wave velocity versus the percentage of melt inclusions[4] in the mantle at 10 km depth. The original Vs in the mantle is assumed to be 4.29 km/s. The different dashed colored lines indicate various aspect ratios (*r*) ranging from 1 to 100, representing the shape of ellipsoid melt inclusion (see inset red ellipse)[4]. *r* = 1 corresponds to spherical inclusions and *r* = 100 represents thin films. The red patch indicates the estimated melt fraction (-0.2–7%) based on a decrease of 0.26 km/s in the S-wave velocity. **b** P-wave velocity as a function for melt (rock) fraction assuming an initial Vp of 7.4 km/s in the mantle, with different P-wave velocities of the melt inclusions varying from 1 to 100 km/s (see legend) for different aspect ratios (*r* = 1, solid lines; *r* = 100, dashed lines). For an increase of 0.45 km/s in the P-wave velocity, P-wave velocities of melt inclusion vary from 10 to 100 km/s and the melt fraction would be 9.4–24.2%. **c** Histograms show the time differences between S- and P-wave phases on each station from Koulakov et al.[1]. The gray columns show the cumulative percentage. **d** Modified Wadati diagram using the method from ref. [10]. The green dots indicate the time differences of P-wave phases (*Pi–Pj*) versus those of S-wave phases (*Si–Sj*) of each station pair (*i,j*) for each event[10]. The dashed black line indicates the estimated Vp/Vs ratio using the full dataset from Koulakov et al.[1], -1.14. The dashed magenta lines mark the different Vp/Vs ratios.

network[1]. In this case, the resolved area in the tomography (25–30 km)[1] is mostly constrained by the earthquakes from short distances, reducing the reliability of deep structures due to the lack of large-offset ray paths.

In addition, we plot the modified Wadati diagram (Fig. 1d) using the station-pair time difference computation[10]:

$$\frac{t_{Si} - t_{Sj} + \Delta t_{water}}{t_{Pi} - t_{Pj} + \Delta t_{water}} = \frac{\frac{D_i}{Vs} - \frac{D_j}{Vs}}{\frac{D_i}{Vp} - \frac{D_j}{Vp}} = \frac{Vp}{Vs} \tag{2}$$

where for a station pair (*i* and *j*), $\Delta t_{water}$ is the travel time difference in the water, $t_{(Pi,Pj)}$ and $t_{(Si,Sj)}$ are travel times for P- and S-waves, respectively, and *(Di, Dj)* are the hypocentral distances. In the study area, the bathymetric data show that water depth differences at all stations will be less than 1 km ($\Delta t_{water} < 0.7$ s), which will not influence Vp/Vs ratios substantially (Fig. 1d). Our obtained Wadati diagram (Fig. 1d) shows that the Vp/Vs ratio is -1.14, much lower than that in the normal oceanic crust and mantle (1.7–2.0), which is evident when time differences are >4 s. We suggest that the picked sP-waves are actually PsP-waves, the main P-waves that have traveled in the crust and mantle, converted to

S-waves at the basement-sediment interface, traveled in the low-velocity unconsolidated sediments as S-waves, and then converted to P-waves at the sediment-water interface. The picked sP-waves on stations G8530-G8533 have much smaller amplitudes than those on other stations (G8510-G8513; G8520-G8523) (See Supplementary Fig. S1 from Koulakov et al.[1]), and we suggest that this discrepancy is likely due to the erroneous identification of S-wave phases. A low S-wave velocity of -200 m/s in the sediments will result in a delay of 0.5 s per 100 m of sediment thickness[11]. Only a 500-m-thick unconsolidated sediment layer can result in an S-wave delay of 2.5 s. As a consequence, the small time differences (sP-P) (<2.5 s) (Fig. 1c) are possible due to large S-wave delays in the unconsolidated sediment layer[11,12]. Although no seismic data are available directly above the volcano, seismic reflection/refraction results[13,14] and sidescan data[15] from other parts of the Gakkel Ridge reveal thick sediments in the axial valley, suggesting that the S-wave delays could indeed have been caused by the presence of thick sediments. As a result, the earthquake hypocenter locations and the tomographic velocity models would be erroneous, and therefore the Koulakov et al.[1] proposal of the presence of volatiles-rich magma reservoir beneath the volcanoes, the low-degree mantle melting and degassing in the mantle would not be valid.

## Data availability

All data generated and analyzed during this study are included in this published article. Source data underlying Fig. 1 are provided as a Source Data file. Source Data are provided with this paper.

## Code availability

The code to reproduce Fig. 1 may be available upon request to the corresponding author.

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

## Acknowledgements

This work was funded by the ISblue project, Interdisciplinary graduate school for the blue planet (ANR-17-EURE-0015), the French government under the program "Investissements d'Avenir", the Regional Council of Brittany (SAD programme), the European Research Council under the European Union's Seventh Framework Programme (FP7/2007–2013), and the ERC Advanced Grant agreement no. 339442_TransAtlanticILAB. This work is IPGP contribution no. 4278.

## Author contributions

Z.Y. performed the computation, analyzed the arrival time data, and wrote the paper. S.C.S. supervised the analysis and interpretation of the seismic data and effective medium theory computations and wrote the paper.

## Competing interests

The authors declare no competing interests.
