## [Peer Review File · Nature Communications]

REVIEWERS' COMMENTS:

Reviewer #1 (Remarks to the Author):

This is A Matters Arising manuscript by Dr. Yu and Dr Singh which is directed to the original paper by Dr. Koulakov et al..

The criticizing authors (MA authors) raised two main issues on this paper. One is the misidentification of seismic phases that should lead to a wrong velocity model. The other is the petrological interpretations on the seismic velocity structure.

I think that the both criticisms are valid and thus will significantly affect the main conclusions in the original paper.

In particular, the first issue is more critical. The authors of the original paper (OP authors) used the coherent data fit as a reason why they think their phase picks are correct.

However, the counterargument by the MA authors is more compelling because seismic waves from deep earthquakes propagate almost vertically and thus causes a trade-off between the depths of hypocenters/conversion points and the velocities above. It seems to me that the OP authors assume that the S-P conversion occurs at the seafloor, but it is very common that it also occurs at the boundary between the basement and sediments on it. Unfortunately, it is very difficult to know where the conversion happens only from seismic waveform. To demonstrate the existence/absence of such a sedimentary layer, the OP authors may need to refer to other data such as seismic reflection images in the study area.

Overall, I agree with the statements by the MA authors and suggest its publication.

Reviewer #2 (Remarks to the Author):

(1) whether you judge that the criticism raised in the comment is valid, Yu et al. point out the misinterpretation of the sP phase by Koulakov et al. Looking at Fig. 1S of the original paper and Yu et al, I would agree the point Yu et al. mentioned. In particular small amplitude of Koulakov et al.'s interpreted sP phase does not explain expected waveform of S-to-P converted wave. However, the data presented by Koulakov et al are only the waveform of one earthquake, and I do not have access to other data. I would need to check other data as well to judge the details. I am actually surprised the original paper does not present any other waveforms in the supplement. With respect to the physical modeling of the V_p/V_s values, the logic of Yu et al. seems reasonable, but both Yu et al. and Koulakov et al. make several assumptions, and there may be different views on the validity of these assumptions.

(2) whether it is likely to be of significant interest to the readers of the original Article. I believe that it is worth to publish the discussion about the point that there may be misinterpretation of the most important data in the original paper. In addition, the authors of the original paper should disclose the seismic waveform data they used.

(3), could it be stated more concisely?
Yu et al. concisely describes the claim.

Responses to the review comments

(The black words show the review comments; the blue words show our responses)

Reviewer: 1

This is A Matters Arising manuscript by Dr. Yu and Dr Singh which is directed to the original paper by Dr. Koulakov et al..

The criticizing authors (MA authors) raised two main issues on this paper. One is the misidentification of seismic phases that should lead to a wrong velocity model.

The other is the petrological interpretations on the seismic velocity structure.

I think that the both criticisms are valid and thus will significantly affect the main conclusions in the original paper.

In particular, the first issue is more critical. The authors of the original paper (OP authors) used the coherent data fit as a reason why they think their phase picks are correct.

However, the counterargument by the MA authors is more compelling because seismic waves from deep earthquakes propagate almost vertically and thus causes a trade-off between the depths of hypocenters/conversion points and the velocities above. It seems to me that the OP authors assume that the S-P conversion occurs at the seafloor, but it is very common that it also occurs at the boundary between the basement and sediments on it. Unfortunately, it is very difficult to know where the conversion happens only from seismic waveform. To demonstrate the existence/absence of such a sedimentary layer, the OP authors may need to refer to other data such as seismic reflection images in the study area.

Overall, I agree with the statements by the MA authors and suggest its publication.

We would like to thank the reviewer for critically reviewing the MA and your positive comments.

Together with reviewer 2, we agree with reviewer 1 that the OP authors should show more seismic waveforms to clarify the phase conversions. Prior to writing the MA, we requested the OP authors to provide us with the data so that we could objectively assess the analysis and results, but we did not get any data.

As far as we know, there are no seismic reflection/refraction profiles in the study area. In addition, the 2021 Chinese JASMINe expedition has observed thick sedimentary reflections by the sonobuoys at the Gakkel Ridge (Ding et al., 2022). We have included this point in the main text, see lines 87, 128-130.

Ding, W. et al., 2022, Submarine wide-angle seismic experiments in the High Arctic: The JASMINe Expedition in the slowest spreading Gakkel Ridge: *Geosystems and Geoenvironment*, v. 1, p. 100076, doi:10.1016/j.geogeo.2022.100076.

Reviewer: 2

(1) whether you judge that the criticism raised in the comment is valid,

Yu et al. point out the misinterpretation of the sP phase by Koulakov et al. Looking at Fig. 1S of the original paper and Yu et al, I would agree the point Yu et al. mentioned. In particular small amplitude of Koulakov et al.'s interpreted sP phase does not explain expected waveform of S-to-P converted wave. However, the data presented by Koulakov et al are only the waveform of one earthquake, and I do not have access to other data. I would need to check other data as well to judge the details. I am actually surprised the original paper does not present any other waveforms in the supplement.

We agree with the reviewer that more seismic waveform data should be enclosed, so that this issue can be clarified.

With respect to the physical modeling of the V_p/V_s values, the logic of Yu et al. seems reasonable, but both Yu et al. and Koulakov et al. make several assumptions, and there may be different views on the validity of these assumptions.

We agree with the reviewer that there could be different views on explaining the V_p/V_s ratios, but one cannot deny the fact that the presence of melt in the mantle will always decrease both V_p and V_s , and will never increase V_p . Therefore, we have changed the title of the MA to 'On increase of P-wave velocity due to melt in the mantle at the Gakkel Ridge' and explain in the text why and what might be the reason.

(2) whether it is likely to be of significant interest to the readers of the original Article.

I believe that it is worth to publish the discussion about the point that there may be misinterpretation of the most important data in the original paper. In addition, the authors of the original paper should disclose the seismic waveform data they used.

Yes, we agree with the reviewer that the authors should publish their seismic waveform data to clarify our concerns.

(3), could it be stated more concisely?

Yu et al. concisely describes the claim.

Thank you very much for your positive comments!